# Engineered Nanobodies for early and accurate diagnosis of dengue virus infection

María Florencia Pavan[1¤a], Cristian Miguel Malnero[2], María Emilia Mora Alvarado[1], María Camila Carzoglio[3], Yesica Paredes Rojas[2], Agostina Bruno[4], Laura Perez Vidakovics[5], Leo Hanke[5¤b], Alejandro Castello[3], Gerald McInerney[5], Cybele Carina García[6], Viviana Parreño[7], Lorena Itatí Ibañez[1]*

1 Laboratorio de Ingeniería de Anticuerpos. Instituto de Química Física de los Materiales, Medio Ambiente y Energía (INQUIMAE), Universidad de Buenos Aires, Consejo Nacional de Investigaciones Científicas y Técnicas (CONICET), Ciudad Autónoma de Buenos Aires, Argentina, 2 Centro de Virología Humana y Animal (CEVHAN), Instituto de Ciencia y Tecnología Dr. César Milstein, Consejo Nacional de Investigaciones Científicas y Técnicas (CONICET), Ciudad Autónoma de Buenos Aires, Argentina, 3 Laboratorio de Inmunología y Virología. Instituto de Microbiología Básica y Aplicada. Departamento de Ciencia y Tecnología. Universidad Nacional de Quilmes. (UNQUI), Buenos Aires, Argentina, 4 Hospital San Vicente de Paul, Laboratorio de Enfermedades Tropicales. Orán, Salta, Argentina, 5 Division of Virology and Immunology, Department of Microbiology, Tumor and Cell Biology, Karolinska Institutet, Stockholm, Sweden, 6 Laboratorio de Estrategias Antivirales. Departamento de Química Biológica (QB), Universidad de Buenos Aires, Consejo Nacional de Investigaciones Científicas y Técnicas (CONICET), Ciudad Autónoma de Buenos Aires, Argentina, 7 Incuinta, Instituto Nacional de Tecnología Agropecuaria (INTA), Instituto de Virología e Innovaciones Tecnológicas, Consejo Nacional de Investigaciones Científicas y Técnicas (IVIT-CONICET), Buenos Aires, Argentina

¤a current address: Novo Nordisk Foundation Center for Protein Research, Department of Cellular and Molecular Medicine, Faculty of Health and Medical Sciences, University of Copenhagen, Copenhagen, Denmark.
¤b current address: Division of Infectious Diseases, Department of Medicine Solna and Center for Molecular Medicine, Karolinska Institutet, Stockholm, Sweden.
* loreitati@gmail.com, itati@qi.fcen.uba.ar

## Abstract

### Background

Dengue virus (DENV), a mosquito-borne flavivirus responsible for dengue disease, has emerged as an escalating global health concern, with cases increasing sharply in recent decades. In Argentina, dengue has transitioned from a sporadic disease to a recurrent epidemic, now affecting 18 of 23 provinces and exposing gaps in diagnostic capacity.

### Methodology and Principal Findings

The DENV genome encodes the non-structural protein 1 (NS1), a key biomarker for early infection detection. Given the limited access to commercially available diagnostic kits within the public health system, we developed a combined ELISA system incorporating Nanobodies designed to target NS1 across all four DENV serotypes as detection antibodies. This system demonstrates excellent discriminative performance (AUC > 0.9), with a diagnostic sensitivity of 93.6% (95% CI: 86.6–97.6%) and

**Data availability statement:** The authors confirm that all data underlying the findings are fully available without restriction. All relevant data are within the paper and its Supporting Information files.

**Funding:** This study was supported by the Agencia Nacional de Promoción de la Investigación, el Desarrollo Tecnológico y la Innovación: PICT 2016-2426 and PICT Start-Up 2019-0016 to LII; and CONVE-2023-100734805-ViroSensar. Ministerio de Ciencia, Tecnología e Innovación to LII. The funders did not play any role in the study design, data collection and analysis, decision to publish, or preparation of the manuscript.

**Competing interests:** The authors have declared that no competing interests exist.

a specificity of 81.1% (95% CI: 70.3–89.3%). The analytical sensitivity showed strong correlation between sera pool dilutions and detected signals, with a limit of detection aligning with reported NS1 concentrations in human samples. While the system exhibits limitations in detecting NS1 from DENV-4, it successfully identified cases in patients five days post-symptom onset who were initially considered epidemiologically negative for dengue infection.

## Significance

Our results underscore the urgent need for accessible, high-precision diagnostic tools in regions facing a surge in dengue outbreaks. Additionally, they highlight the necessity of revising current diagnostic algorithms to enhance the detection of late-presenting cases.

## Author summary

Dengue virus is a major public health concern in Latin America, with recurring outbreaks placing significant strain on healthcare systems. Early detection is crucial for effective patient management and controlling the spread of the virus. This study focuses on the development and optimization of a combined ELISA-based detection system for dengue virus, leveraging Nanobody technology to enhance sensitivity and specificity. Compared to traditional methods, this approach offers a cost-effective and scalable solution, making it particularly valuable for resource-limited settings. The findings presented here underscore the importance of robust diagnostic algorithms that can accurately identify both early and late-stage infections, addressing gaps in current detection strategies. By improving the precision and accessibility of dengue diagnostics, this research contributes to public health initiatives aimed at mitigating the impact of outbreaks across Latin America. The insights gained could support local and regional efforts in establishing stronger surveillance programs and informed policy decisions, ultimately reducing the burden of dengue in affected communities.

## Introduction

Dengue infection, caused by mosquito-borne viruses, poses a significant global public health challenge. The virus's spread is driven by factors such as climate change, global population growth, travel to endemic areas, and inadequate waste management [1]. Outbreaks have been reported in over 121 countries [2,3]. According to the World Health Organization (WHO), reported cases rose from 505,430 in 2000 to over 14 million in 2024 [4]. While many cases are asymptomatic or mild, leading to significant underreporting, the impact of the disease continues to escalate. Since the 1980s, Latin America and the Caribbean have witnessed a significant rise in

dengue cases, increasing from 3.76 million (1980-1999) to 12.68 million (2010-2017) [5]. From 2018 to 2022, over 9.97 million cases and 4,681 deaths were recorded [6]. In 2023, nearly 3 million suspected and confirmed cases surpassed 2.8 million from 2022. By early 2024, Latin America had become the most affected region globally, with approximately 11.9 million confirmed cases.

In Argentina, the epidemiology of dengue has transformed from a sporadic disease to a recurrent public health issue. Initially confined to northern regions, the virus has progressively spread, and now 18 of Argentina's 23 provinces report autochthonous dengue cases [7,8]. The 2009 outbreak was the largest recorded up to that point, with 26,923 confirmed DENV-1 cases across 14 provinces, 10 of which had no prior epidemiological history of the disease [9]. In early 2016, a new outbreak introduced serotype 4 (DENV-4) and surpassed the 2009 epidemic by 49% [9,10]. Dengue cases in Argentina have recently reached unprecedented levels, with 129,500 confirmed cases in 2022/2023, rising dramatically to 566,140 in 2023/2024, and leading to a historic peak in transmission and deaths. In 2024, only 35% of cases were confirmed through laboratory testing, highlighting a significant gap in diagnostic capacity and suggesting underreporting [8]. This underscores the need to strengthen laboratory infrastructure and diagnostic tools to better manage dengue in Argentina.

The dengue virus, part of the Flaviviridae family, has four distinct serotypes: DENV-1, DENV-2, DENV-3, and DENV-4, distinguishable by serological and molecular methods [11]. Its genome, a single-stranded RNA, translates into a polyprotein, yielding three structural proteins (C, E, M) and seven non-structural proteins (NS1, NS2A/2B, NS3, NS4A/4B, NS5) [11]. NS1 is a 46–50 kilodalton (kDa) glycoprotein that displays high amino acid and nucleo-tide homology across multiple flaviviruses [12,13]. Sequence similarity among NS1 proteins from the four dengue virus serotypes ranges between 68% and 80%. In comparison, NS1 shares approximately 54% identity with Zika virus (ZIKV), 51–53% with Japanese encephalitis virus (JEV), and 49–56% with West Nile virus (WNV) [13]. The NS1 protein is assembled into multiple oligomeric forms. The dimeric form, predominantly found within host cells, facilitates interactions with host proteins and plays a critical role in viral replication. According to several reports, secreted NS1 can also exist in both hexameric and tetrameric forms, which contribute significantly to viral replica-tion, immune evasion, and pathogenesis [14,15]. This protein is considered a potent early biomarker for dengue virus detection, given its secretion into the bloodstream during the acute phase of infection. It is detectable from day 1 of fever onset and remains present for up to 7–9 days in most patients, although it has been detected as late as day 18 [16]. High serum levels of secreted NS1 (sNS1), ranging from 0.4 µg/mL to 10 µg/mL, are detectable early in infections [12,17–19] and may correlate with disease severity [20,21], making it a valuable marker for diag-nosis and prognosis.

In regions where dengue is endemic, efforts to eradicate *Aedes aegypti* mosquitoes, the primary vectors for dengue and other flaviviruses, require enhancement through comprehensive and multifaceted programs [22,23]. Advancements in therapeutic strategies, including the development of vaccines and antivirals, are anticipated to play a crucial role in con-trolling the disease. Two tetravalent live attenuated vaccines—CYD-TDV (Dengvaxia) and TAK-003 (Qdenga)—have been authorized for use in several countries [24,25]. Although no specific antiviral treatment for dengue has been approved to date, promising therapeutic options targeting viral proteins and host factors are currently under development [26,27].

Dengue symptoms are often nonspecific and can be easily confused with other flavivirus infections or other viruses such as influenza or coronavirus [28]. Thus, using at least two techniques to accurately determine DENV infection is recommended [29,30]. Accurate diagnosis is crucial for pre-vaccination screening and early detection, which is vital to prevent disease spread and ensure effective management. Recent advancements in diagnostic tools include traditional virus isolation, which remains time-consuming and costly, as well as RNA detection methods such as RT-PCR and LAMP, which require specialized laboratories and trained personnel [11,31,32]. Serological testing for IgM, IgG, and IgA anti-bodies, commonly used in remote areas, does not identify DENV serotypes and may cross-react with other flaviviruses [32]. Emerging technologies include micro- and nanotechnologies, immunosensors, lateral flow immunoassays, and

microarrays [33–38]. NS1 antigen detection by ELISA is rapid and straightforward but imported commercial kits are expensive, which may be prohibitive for certain health systems, especially in developing countries [17,39].

Single domain antibodies, discovered over 30 years ago in camelids, include two IgG isotypes (IgG2 and IgG3) lacking the light chain and the CH1 domain [40,41]. The variable fragment of these heavy-chain-only antibodies is called VHH or Nanobody (Nb) [42]. VHH, the smallest functional antigen-binding fragment, binds specifically and with high affinity to target antigens. It can be modified for better thermal stability or to create multivalent molecules. These antibody fragments have been successfully used to inhibit viral infections and detect various pathogens [43,44].

Given the need for innovative, early infection detection systems and the high costs of imported commercial kits, especially with the global rise in dengue cases, here we selected Nbs that specifically recognize the NS1 protein of all four DENV serotypes. These Nbs were later modified to enhance their properties as capture or detection antibodies and were used to develop a validated combined Nanobody ELISA (cNb-ELISA) test for diagnosis of dengue virus infection.

## Methods

### Ethics statement

Anonymized human serum samples (2016-2019) were obtained from the Laboratory of Tropical Diseases, San Vicente de Paul Hospital, Orán, Salta, Argentina. The protocol was approved by the Provincial Commission on Health Science Research (COPICSA), Ministry of Health of Salta, under reference number 321-196940/18, and complies with the Declaration of Helsinki. Sample information has been previously published [45].

### NS1 protein production

NS1-containing supernatant from different serotypes was obtained by infecting C6/36 insect cells with DENV-1 (Hawaii), DENV-2 (New Guinea C), DENV-3 (Philippines/H87/1956), and DENV-4 (H241) at a multiplicity of infection (MOI) of 1. Five days post-infection (p.i.), the supernatant was harvested, clarified by centrifugation, and concentrated four times by ultrafiltration (30 kDa cut-off, Vivaspin). Culture media was then exchanged to phosphate buffer saline (PBS) and used to immunize a llama.

For cross-reactivity assays, Vero CCL-81 cells were infected with DENV-1 to -4, Chikungunya virus (CHIKV), and Yellow Fever virus (YFV) at a MOI of 0.1. Cell culture supernatants were collected between days 3 and 7 p.i., clarified by centrifugation, filtered, and stored at −80 °C until use. Viral titers were quantified by plaque assay and ranged from $1 \times 10^6$ to $1 \times 10^7$ plaque-forming units (PFU) per milliliter. Viral strains were kindly provided by Dr. Alicia Mistchenko from the Laboratory of Virology, Hospital de Niños Dr. Ricardo Gutiérrez, Buenos Aires, Argentina.

Recombinant NS1 was generated through PCR amplification, followed by cloning using either restriction enzyme-based or seamless approaches. Primers are listed in S1 Table. The cDNA template was prepared from RNA extracted from C6/36 insect cells infected with the dengue virus strains previously mentioned. Two groups of constructs were used to clone NS1 from all four DENV serotypes: a pCAGGS-ss vector containing the human serum albumin secretion signal and a His-tag [46], and a modified pFUSE vector (invivoGene) with an IL2 secretion signal, a Flag-tag at the N-terminus, and a His-tag at the C-terminus. Recombinant NS1 protein from Zika virus (MR766) was also cloned into the same vectors. For protein extraction, vectors were transfected into HEK-293T cells using polyethyleneimine (PEI, polyAR). The supernatant was collected 72 hours post-transfection, adjusted with phosphate buffer or Tris-NaCl, and purified by IMAC using a Ni-NTA resin. DENV-2, cloned into the pCAGGS-ss vector, was expressed in Vero cells for biopanning experiments. Transfection was carried out on a 96-well plate using Lipofectamine 2000. Each well was seeded with $1.5 \times 10^4$ Vero cells. After 24 hours, 60 ng of plasmid DNA, diluted in 50 µL of Opti-MEM, was mixed with 0.2 µL of Lipofectamine 2000 (also diluted in 50 µL of Opti-MEM). The mixture was incubated at room temperature for 25 minutes to allow complex formation. Transfection complexes were then added to the cells, which were incubated for 48 hours at 37°C in a 5% $CO_2$ incubator to enable protein expression.

## Selection of NS1-specific Nanobodies

Animal immunizations were approved by the Ethical Committee of the Faculty of Veterinary Sciences, University of Buenos Aires (updated protocol 2022/23). A llama was immunized with five injections of 250 µL of purified supernatant containing NS1 of each DENV strain. Complete Freund's adjuvant was used for the first immunization and Incomplete Freund's for boosters on day 28, 41, 58, 83 and 90. Serum samples were collected before the first immunization and two weeks after each injection.

Antibody titers were monitored by ELISA using 96-well plates coated with 100 ng of recombinant NS1 proteins. Plates were blocked with 3% skimmed milk in PBST (PBS + 0.05% Tween 20), then incubated with five-fold diluted serum. Horseradish peroxidase (HRP)-linked anti-llama IgG (Bethyl Laboratories, 1:15,000) was added, and the reaction was developed with 3,3',5,5'-Tetramethylbenzidine (TMB) and read at 450 nm. The endpoint titer was considered as the highest sample dilution with an absorbance at 450 at least double that of the pre-immune serum sample.

Four days after the final immunization, 150 mL of blood was collected. Peripheral blood lymphocytes were isolated using Ficoll Paque Plus filled Leucosep tubes, and RNA was extracted using RNeasy Midi columns. The VHH-gene library was constructed following the protocol published by Pardon *et al.* [47].

Biopanning was conducted in 96-well plates using either recombinant NS1-1 (cloned from DENV-1, 0.2 µg/well), or Vero cells transfected with pCAGGS-ss-NS1-2 construct (cloned from DENV-2, 60 ng/well) and fixed with acetone/ethanol 24 hours post-transfection. Welles prepared for both selection strategies were incubated at room temperature for 1 hour with 2% skimmed milk in PBST. Recombinant phages were then subjected to three successive rounds of counter-selection, each involving a 15-minute incubation at room temperature of 200 µL of phage suspension (at a concentration of $10^{12}$ phages/mL) in wells containing protein extracts derived from non-transfected HEK293T or Vero cells, as appropriate. Counter selection steps were included to diminish non-specific phage binding, as the llama was immunized with a supernatant containing several irrelevant proteins. After the final counter-selection step, 100 µL of the collected phage suspension was transferred to the selection wells containing either the recombinant NS1-1 protein or transfected Vero cells. The remaining 100 µL was added to wells containing protein extracts from non-transfected HEK293T or Vero cells, serving as negative controls. Plates were incubated for 2 hours at room temperature. Non-specifically absorbed phages were removed by increasing the number of washes with PBST after each round of panning. Bound phages were eluted a two-step elution process, using trypsin solution (T-elution) and exponentially growing TG1 cells (C-elution) as previously published [48]. The first elution step involved the incubation for 30 minutes at room temperature with 100 µL per well of TPCK-treated trypsin (0.25 mg/mL in PBS). Enzymatic activity was quenched by adding 5 µL per well of AEBSF protease inhibitor (4 mg/mL). A second elution step consisted of the incubation at 37°C for 30 minutes with 200 µL of exponentially growing E. coli TG1 cells. Serial dilutions of infected bacteria were seeded to control antigen-specific phage enrichment and to select single colonies for further characterization.

## Screening and characterization of NS1-specific Nanobodies

A total of 192 individual colonies from the second and third rounds of panning, conducted using either T- or C-elution protocols with recombinant NS1 protein or transfected Vero cells, were analyzed by ELISA. Nanobodies were extracted after induction with 1 mM IPTG for 6 hours at 37°C. ELISA plates were coated overnight at 4°C with 0.2 µg/mL of recombinant NS1-1 or NS1-2 proteins. The next day, plates were blocked with 3% skimmed milk in PBST, then incubated with 100 µL of periplasmic extract for 2 hours at room temperature. Plates were washed three times, and 50 µL/well of HRP-conjugated anti-HA antibody (1:1,500) was added for 1 hour to detect specific binding. The reaction was developed with TMB and read at 450 nm.

Ninety-six clones (1–48 selected using recombinant NS1-1 protein and 49–96 using a transfected DENV-2 NS1 plasmid) were amplified via PCR with primers listed in S1 Table. The PCR fragments were then analyzed by RFLP, utilizing the HinfI restriction enzyme for digestion. Samples were run on a 2.5% agarose gel to determine fragment patterns.

After sequencing, genetic similarity was analyzed using a Neighbor-joining tree with 1000 bootstrap replicates (MEGA version 11) [49]. The sequence logo was plotted using WebLogo3. Selected Nb sequences were subcloned into the pHen6 vector using NcoI and NotI restriction enzymes, and WK6 cells were transformed for Nb expression and scale-up.

## Nanobody modifications

To use the selected Nbs for NS1 protein capture or detection in the enzyme immunoassay, we made several modifications. Two polystyrene (PS) binding peptide sequences, rich in tryptophan (VHWDFRQWWQP, called PSW) and rich in arginine (RIIIRRIRR, called PSR) were selected to modify Nbs for NS1 capture [50,51]. Nanobody sequences were amplified by PCR using specific primers and cloned into a vector with the PSW or PSR sequence at the N-terminal. The modified Nbs with the PSR sequence encountered expression and purification issues during protein production scaling; therefore, in this study, only the Nbs modified with the PSW sequence were used. Bivalent Nbs were prepared by cloning the same Nb separated by a 20 Glycine-Serine linker $(G_4S)_4$. To produce the Nb Fc fusions, Nb sequences were subcloned into a modified pFUSEss eukaryotic expression vector (InvivoGen). This vector incorporates an IL-2 secretion signal and the murine IgG2a fragment, comprising the hinge, CH2, and CH3 domains.

Nanobodies for NS1 detection were modified by biotinylation applying two different techniques. In one, Nb sequences with an AviTag and His-tag at the C-terminus were cloned into a pET22-based vector with the BirA biotin ligase sequence under the T7 promoter (Addgene plasmid # 100817) [52]. The second method involved the use of a sortase A reaction after adding the LPETG target sequence at the C-terminus of each Nb via Gibson Assembly (Addgene plasmid # 75144) [53,54]. Nbs for detection were also modified by adding an Fc tag followed by the HRP coding sequence. The CH1 domain was deleted from a pFUSE vector, the stop codon was removed, and the HRP sequence was cloned into the same ORF. All constructs were made using In-Fusion seamless cloning with primers listed in S1 Table.

## Expression and purification of Nanobodies

Production and purification of the selected clones (n = 11), as well as Nbs PSW and bivalent Nbs, were carried out at 37°C in Terrific Broth (TB) medium supplemented with 100 µg/mL ampicillin and 0.1% glucose. Nanobody expression was induced with 1 mM IPTG for 16 hours at 28°C. The proteins were purified from periplasmic extracts by IMAC using a Ni-NTA resin and eluted with 300 mM Imidazole.

To biotinylate Nb with BirA, plasmids were transformed into BL21 bacteria, expression was induced with IPTG at 28°C. The cell culture was centrifuged, and the pellet was freeze-thawed in the presence of 50 µM D-biotin (Sigma), to rupture cell membranes and allow interaction between Nbs and biotin ligase. Bacterial lysate was incubated at 37°C for 30 minutes to enhance biotinylation yield [55]. Biotinylated Nbs were then purified by Ni-NTA resin using chromatography columns. In the case of biotinylation with sortase A, purified Nbs (50-70 µM) were incubated with 5 µM sortase A and 5 µM Biotin-PEG-NH2 in a reaction buffer (150 mM NaCl, 50 mM Tris pH 7.5, 10 mM $CaCl_2$) for 4 hours on ice. Unreacted Nbs and sortase A were removed using Ni-NTA resin, while excess nucleophile was eliminated with a PD-10 desalting column [56].

Nanobodies Fc and Nbs Fc-HRP fusions were transiently expressed in HEK-293T. The supernatant containing recombinant proteins was harvested 72 hours post-transfection and purified by Protein G affinity chromatography. Fractions containing the recombinant proteins, as judged by the SDS-PAGE analysis, were pooled, dialyzed against PBS, and stored at −80°C.

## Evaluation of Nanobodies properties for capture and detection

To assess the ability of the different Nbs to capture NS1, 96-well ELISA plates were coated with 10 µg/mL of Nbs overnight at 4°C. Plates were blocked for 1 hour at room temperature with 5% equine serum, 3% BSA, 1% BSA, or 3% milk in PBST. After three PBST washes, 100 ng/well of recombinant NS1-2 was added and incubated for 1 hour and 30 minutes

at room temperature. Plates were later incubated with anti-Flag HRP-conjugated monoclonal antibody (1:10,000) for 1 hour and 30 minutes at room temperature to detect NS1. An irrelevant Nb (specific for the receptor-binding domain of SARS-CoV-2) was used as a negative control. The reaction was developed with TMB and read at 450 nm. To compare the capture efficiency of different Nbs, microplates were coated with a serial three-fold dilution of Nbs, starting from 800 nM, and 50 ng/well of recombinant NS1-2 was used as antigen. Detection of NS1 was assessed as described in the former assay. These experiments were performed in triplicates. All Nbs were also evaluated as capture antibodies—both in their monovalent form and as Fc-fusion constructs—using clinical serum samples, following the establishment of the ELISA protocol (see Optimization of the Nanobody-based ELISA).

To evaluate the NS1 detection capacity of selected Nbs, MaxiSorp 96-well ELISA plates were coated overnight at 4°C with 1 µg/mL recombinant NS1-2 protein in carbonate/bicarbonate buffer (pH 9.6). Plates were blocked with 10% skimmed milk in PBST for 1 hour at 37°C. Purified Nbs were adjusted to a starting concentration of 1 µM, then serially diluted 5-fold and added to the plates for 1 hour at 37°C. A monoclonal Anti-Dengue virus NS1 protein antibody (BEI resources NR-10121) and an irrelevant Nb were also serially diluted in a 5-fold dilution series and used as positive and negative controls, respectively. A commercial HRP-conjugated anti-VHH antibody (1:8,000) was added. The reaction was developed with TMB and read at 450 nm. The $EC_{50}$ was estimated from triplicates measurements by a four-parameter log-logistic regression model. Similar conditions were applied when Nbs coupled to HRP were used as detection antibodies.

To evaluate potential epitope overlap among Nb14, Nb48, and Nb66, a competitive ELISA was conducted using unlabeled monovalent Nbs as competitors and their corresponding Fc-fusion formats as detection probes. Ninety-six-well ELISA plates were coated overnight at 4°C with NS1-2 protein (2 µg/mL), followed by blocking with 3% (w/v) skim milk in PBST for 1 hour at room temperature. Serial two-fold dilutions of the unlabeled Nbs (starting at 60 µg/mL) were incubated with a fixed concentration of Nb-Fc (0.4 µg/mL) and added to the wells for 1 hour. After washing, bound Nb-Fc was detected using an HRP-conjugated anti-mouse antibody (1:5,000 dilution). Signal development was carried out using TMB substrate as previously described.

## Optimization of the Nanobody-based ELISA

Optimal concentrations of capture and detection antibodies were determined by a chessboard titration experiment. Microplates were coated overnight at 4°C with serial two-fold dilutions of Nb14 Fc, starting from either 4 or 2 µg/mL. Plates were blocked with 1% BSA in PBST for 1 hour at 37°C. After three PBST washes, 50 ng/well of recombinant NS1-2 was added and incubated for 1 hour and 30 minutes at 37°C. Each capture antibody concentration was tested against six detection antibody concentrations (Nb48 or Nb66, in Fc-HRP or biotinylated format): 1, 5, 10, 15, 20 and 25 µg/mL, incubated for 1 hour and 30 minutes at 37°C. For biotinylated Nbs, an additional incubation with streptavidin-HRP (1:5,000) was performed for 1 hour at 37°C. The reaction was developed with TMB and read at 450 nm. The assay was run with duplicate wells for each sample.

## Production of a rabbit-derived polyclonal anti-NS1 serum

Experiments with rabbits were conducted in collaboration with researchers from the Universidad Nacional de Quilmes (approved protocol 001/16, updated protocol 004/24). A rabbit was subjected to a five-dose immunization protocol over a four-month period. The first two immunizations (day 0 and day 28) used 200 µg of pCAGGS NS1-1, NS1-3, and NS1-4 plasmids. A protein booster (30 µg of recombinant NS1-1, NS1-3, and NS1-4) was administered at day 56. The final two immunizations (day 91 and day 119) used 200 µg of pCAGGS NS1-1 plasmid. Blood was collected, and serum was extracted and purified to isolate IgG produced in response to immunization. Purified rabbit IgG titration was performed in triplicate via ELISA on plates coated with 2 µg/mL of each NS1 protein. Specific recognition was detected using an anti-rabbit HRP-coupled antibody (1:3,000).

## Development and statistical validation of a combined Nanobody ELISA assay

To assess the accuracy and performance of the combined Nb ELISA (cNb-ELISA), serum samples from patients, categorized as NS1-positive using PLATELIA DENGUE NS1 Ag (BioRad) (n=94) or epidemiologically negative (n=74), were analyzed. Each assay plate included two internal controls: a pool of highly reactive positive sera and a low-reactivity negative serum. Microplates were coated with 10 µg/mL of anti-NS1 purified rabbit IgG in carbonate/bicarbonate buffer and incubated overnight at 4°C. The next day, plates were blocked with 1% BSA in PBST for 1 hour at 37°C. After four PBST washes, 25 ng/well of recombinant NS1 from the four DENV serotypes and ZIKV in a negative serum matrix, a pool of positive sera (diluted 1:2), or individual patient samples (diluted 1:2) were added and incubated for 1 hour and 30 minutes at 37°C. After washing, 5 µg/mL of biotinylated Nb66 was added and incubated for 1 hour and 30 minutes at 37°C. An additional incubation with streptavidin-HRP (1:5,000) was performed for 1 hour at 37°C. The reaction was developed with TMB and read at 450 nm. The cNb-ELISA protocol was further employed to assess the specificity of the detection system against other circulating viruses. For this purpose, supernatants from Vero cells infected with DENV serotypes, ZIKV, CHIKV, and YFV were analyzed.

## Statistical methods included in the combined Nanobody ELISA validation

Absorbance values at 450 nm were used to construct a receiver operating characteristic (ROC) curve to evaluate the assay's diagnostic performance (cut-off and diagnosis sensitivity and specificity). The area under the ROC curve (AUC) and its 95% confidence interval (CI) were calculated to determine the assay's discriminative ability. The optimal cut-off value was established based on the Youden index and corroborated using the MaxSpSe method. Diagnostic sensitivity and specificity were then computed at the determined cut-off, with their respective 95% CIs. Data analysis was performed using the easyROC package in RStudio (version 3.6.1).

Concordance or agreement between reference samples (positive detected by PLATELIA DENGUE NS1 Ag (BioRad) assay performed at Orán Hospital, and negative assigned by an epidemiological algorithms) and our developed assay was assessed by the calculation of the Cohen's kappa index [57]. Samples giving discordant results were tested using Dengue virus NS1 Ag (DIA.PRO), according to the provider protocol.

To determine the limit of detection (LOD), two-fold serial dilutions of recombinant NS1 of the indicated serotypes were spiked into 100% human sera from non-infected patients. Every sample was measured in triplicates and data was fitted to a 4-parameter logistic equation:

$$Y = Bottom + (Top - Bottom)/(1 + 10\hat{}((LogEC50 - X) * HillSlope))$$

The LOD was calculated based on the standard deviation of the response (Sy) of the curve and the slope of the calibration curve (S) at levels approximating the LOD according to the formula: LOD = 3.3*(Sy/S).

The analytical sensitivity of the cNb-ELISA was evaluated using a pooled set of positive sera, serially diluted twofold from 1:2 to 1:256. Absorbance values at 450 nm were plotted against the base-10 logarithm of the dilution factor, and nonlinear regression analysis was performed. The linear range of the assay was identified by linear regression analysis, evaluating the significance of the linear relationship through the calculation of the coefficient of determination ($R^2$), along with the corresponding slope and intercept values to ensure the robustness of the system.

## Results

### Production and characterization of DENV NS1 specific Nanobodies

Heavy-chain-only antibodies specific to the NS1 protein were raised in a llama immunized with purified supernatants obtained from C6/36 insect cells infected with four DENV serotypes. After five immunizations, anti-DENV NS1 specific antibody titers increased (S1A Fig), and an immune library of $1 \times 10^8$ transformants was constructed using peripheral

blood lymphocytes. Colony PCR on randomly picked colonies identified the presence of inserts in 78.5% of transformants (S1B Fig). Recombinant and transfected NS1-1 and NS1-2 specific binders were selected through three rounds of panning using two elution strategies established in our laboratory [48]. Although all dengue serotypes have been reported to circulate in Argentina, we selected serotypes 1 and 2 for panning, as they have shown the highest frequency of circulation in the country in recent years [58–61]. Cross-reactivity of Nbs to NS1-1 and NS1-2 was analyzed by ELISA using wells coated with recombinant NS1 proteins. As shown in S2 Fig, most clones exhibited varying degrees of reactivity against both NS1-1 and NS1-2. Ninety-six clones were subjected to RFLP sequence analysis, revealing 18 different restriction patterns (S3 Fig). Plasmids were sequenced and 11 distinct Nbs (Nb5, Nb7, Nb14, Nb22, Nb34, Nb40, Nb48, Nb51, Nb-4, Nb66, and Nb70) with conserved residues in the framework regions (FRs) and high variability in the complementarity-determining regions (CDRs) were identified (S4 Fig). Phylogenetic and bioinformatic analysis of the selected Nbs sequences revealed three main groups, providing insights into their germline origin (S5 Fig).

Six Nbs were selected for further characterization based on phylogenetic analysis: four representatives of the identified phylogenetic groups (Nb5, Nb22, Nb48, and Nb66) and two ungrouped Nbs (Nb14 and Nb40). Despite numerous attempts, Nb70 and Nb40 failed to achieve sufficient expression levels for downstream assays and were therefore excluded from most experiments.

The specificity and affinity of the selected Nbs was tested by ELISA against recombinant NS1 proteins from all DENV serotypes and ZIKV. All Nbs demonstrated specificity for DENV NS1 proteins, with varying degrees of relative affinity depending on the Nb and serotype. None of the Nbs showed reactivity toward ZIKV NS1 (S6 Fig). Notably, all Nbs, except Nb22, exhibited single digit nanomolar $EC_{50}$ values, indicating strong binding to DENV NS1 (Table 1). Nb14, Nb40, and Nb66 displayed the highest relative affinities, highlighting their potential as diagnostic reagents for assay development. A monoclonal Anti-NS1 protein antibody and an irrelevant Nb were included as positive and negative controls, respectively.

To identify shared epitopes among the different Nbs under investigation, we performed a competition assay. In each reaction, we combined Fc-tagged Nbs—whose binding was detected using an HRP-conjugated anti-mouse antibody—with monovalent Nbs. This approach allowed us to determine that Nb48 and Nb14 compete for the same binding site on the NS1 protein, whereas the epitope recognized by Nb66 is located elsewhere on the protein (S7 Fig).

The selected Nbs were analyzed by ELISA to identify the best Nb for NS1 antigen capture. Based on binding data (S6 Fig and Table 1), Nb14 demonstrated strong suitability for NS1 capture and was used to establish assay conditions. Optimal assay performance was determined by comparing two microplate brands and three blocking reagents (3% and 1% BSA, 5% equine serum, and 3% skimmed milk). Results indicated that 1% BSA was the most effective blocking reagent for recombinant NS1 capture when paired with Greiner microplates, exhibiting the highest P/N ratio (Fig 1A).

**Table 1. $EC_{50}$ values of selected Nanobodies against DENV NS1 proteins.**

| Nb | $EC_{50}$ (nM) | | | |
|---|---|---|---|---|
| | NS1-1 | NS1-2 | NS1-3 | NS1-4 |
| Nb5 | 3.031 | 0.146 | 0.371 | 1.603 |
| Nb14 | 0.056 | 0.047 | 0.063 | 0.064 |
| Nb22 | 35.14 | 22.51 | 52.51 | 41.84 |
| Nb40 | 0.112 | 0.034 | 0.069 | 0.074 |
| Nb48 | 4.549 | 1.338 | 44.59 | 25.16 |
| Nb66 | 0.071 | 0.076 | 0.112 | 0.137 |
| IrrNb | ND | ND | ND | ND |
| mAb | 23.43 | 1.71 | 9.76 | 3.33 |

ND: not detected

Under optimized conditions, the capture capacities of nine Nbs for NS1-2 were evaluated at two concentrations using an anti-Flag HRP antibody for detection. Nb14 demonstrated the highest capture efficiency, with absorbance values close to 1 at both concentrations (Fig 1B). Nanobody 48 also showed the ability to capture NS1-2, though with lower absorbance values compared to Nb14. These findings suggest that Nb14 and Nb48 are promising candidates as capture antibodies for the diagnostic assays.

### Development of capture and detection Nb-based tools for dengue diagnosis

To address the high quantity of Nb required for effective capture and the critical role of proper orientation in improving detection, Nb14 was engineered to enhance capture efficiency while reducing the necessary amount of protein. Key modifications included the incorporation of a polystyrene-binding sequence (Nb-PSW) to improve microplate immobilization, the construction of a bivalent Nb to enhance avidity, and the fusion of the Fc domain (Nb Fc) to combine the benefits of enhanced avidity and improved immobilization through Fc-mediated orientation on plastic surfaces. Additionally, biotinylated Nbs and Fc-HRP fusions of Nb48 and Nb66 were developed for Nb-based detection tools.

NS1 capture efficiency was assessed after serial dilutions of each engineered Nb. The capturing Nb14 Fc showed a 63-fold increase in ELISA compared to the monovalent Nb14, with the lowest $EC_{50}$, confirming its binding capacity for NS1. The bivalent Nb also showed significantly improved capture capacity compared to the monovalent formats. In contrast, the monovalent Nb and Nb PSW displayed similar titration curves and $EC_{50}$ values, indicating that the addition of the polystyrene-binding sequence did not significantly improve capture efficiency (Fig 2, Table 2).

Additionally, due to scale-up challenges, the PSR-modified Nbs were not tested in our assay. While several strategies exist to address low expression yields—such as fusion to Maltose Binding Protein or protocols tailored for recalcitrant Nbs—we opted not to pursue yield optimization [62–64]. This decision was based on the lack of significant improvement in capture performance following the addition of the PSW polystyrene-binding peptide.

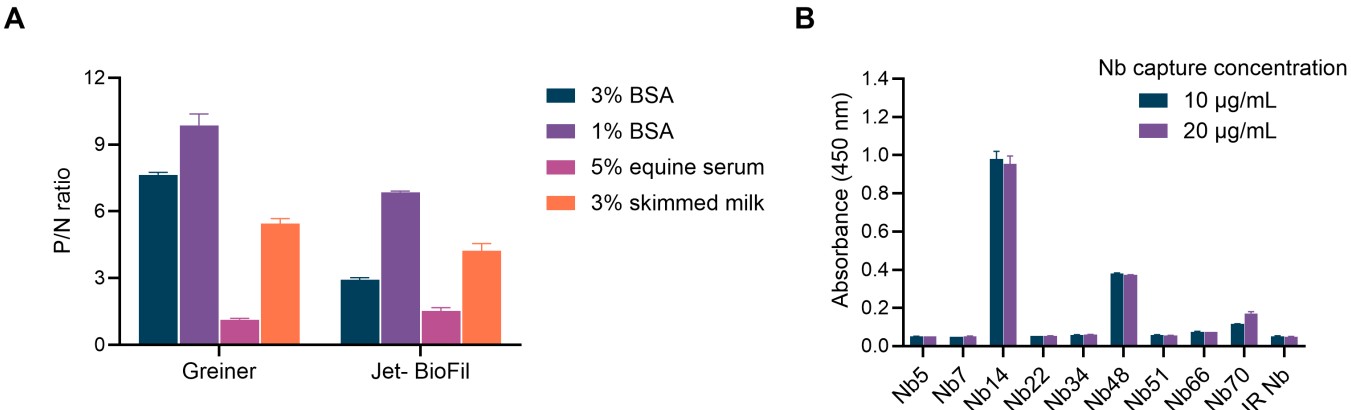

**Fig 1. Determination of optimal conditions for NS1 capture.** A-Various blocking agents and two distinct ELISA microplates were evaluated. The P/N ratio for NS1-2 capture indicated that 1% BSA, combined with Greiner microplates, yielded the highest P/N ratio. B-The ability of the Nbs under study to capture the NS1-2 protein was assessed using two different concentrations (10 and 20 µg/mL). Detection was performed using an anti-Flag HRP antibody. Nanobody 14 and Nb48 exhibited a stronger capture capacity compared to other tested Nbs.

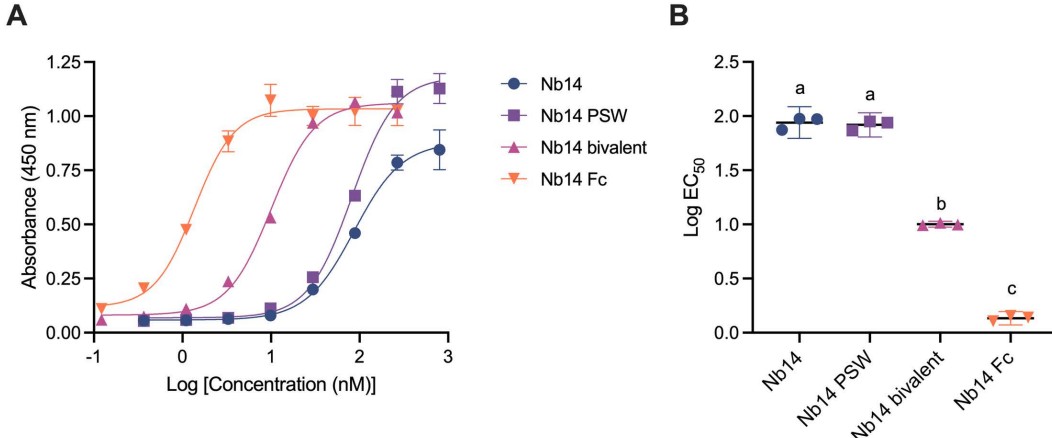

**Fig 2. Comparison of the NS1 capture capacity of modified Nb14.** A-Titration curves of unmodified Nb14, Nb14 PSW, bivalent Nb14, and Nb14 Fc. Absorbance at 450 nm was plotted as a function of Nb concentration. B-Analysis of log $EC_{50}$ for each Nb14 format, indicating significant differences in capture capacity. Letters (a, b, c) indicate groups with significant differences (one-way ANOVA, Tukey's multiple comparison test, $p < 0.01$).

**Table 2. $EC_{50}$ values of modified Nb14.**

| Format | $EC_{50}$ (nM) |
|---|---|
| Nb14 | 86.66 |
| Nb14 PSW | 81.59 |
| Bivalent Nb14 | 10.03 |
| Nb14 Fc | 1.36 |

## Identification of the optimal Nanobody pair for dengue diagnosis

Nanobody 14 Fc was designated as the capture antibody for the Nb-ELISA development, and a chessboard titration was conducted to optimize both capture and detection antibody concentrations. Nanobody 48 and Nb66, representing two distinct Nb families, based on phylogenetic and germline analyses, were used as detection antibodies in Fc-HRP fusion (S8 Fig) and biotinylated (S9 Fig) formats.

Optimization applying HRP-fused Nbs revealed that lower concentrations of Nb14 Fc as the capture antibody improved signal strength, likely by increasing availability for NS1 binding and reducing surface saturation (S8 Fig). For Nb14 Fc/Nb48 HRP pair, the highest P/N ratio was achieved with 4 µg/mL capture antibody and 10 µg/mL detection antibody (S8B Fig). Similarly, for Nb14 Fc/Nb66 HRP, the optimal signal was observed with 4 µg/mL capture antibody and 20 µg/mL detection antibody, with a P/N ratio close to 6 (S8D Fig). Based on these results, the first system for NS1 detection, named system 1, was proposed: 4 µg/mL Nb14 Fc as capture antibody, 10 µg/mL Nb48 HRP as detection antibody, and 1% BSA as the blocking reagent.

Two biotinylation strategies were employed to modify detection antibodies. The biotinylation efficiency using the sortase A system was significantly higher than that of the AviTag system (S10 Fig). Therefore, Nbs labeled with biotin via the sortase A system were used in the subsequent determinations. When utilizing these modified Nbs, a trend observed with HRP-conjugated Nbs emerged, with higher absorbance values achieved at even lower Nb14 Fc concentration (2 µg/mL; S9 Fig). The highest P/N ratio for the Nb14 Fc/Nb48 biotin pair was obtained using 2 µg/mL capture antibody and 10 µg/mL detection antibody (S9B Fig). For Nb14 Fc/Nb66 biotin pair, the optimal combination was 2 µg/mL capture antibody and 5 µg/mL detection antibody (S9D Fig). These results demonstrate that biotinylated Nbs enhance signal amplification

and improve the sensitivity of the sandwich ELISA. Therefore, another system, named system 2, was proposed: 2 µg/mLNb14 Fc as the capture antibody, 5 µg/mL Nb66 biotin as the detection antibody, and 1% BSA as the blocking reagent.

## Evaluation of NS1 Detection Systems in a Serum Matrix

The two optimized systems were tested for their ability to detect recombinant DENV and ZIKV NS1 proteins in a serum matrix (serum from a DENV NS1 negative patient) that does not exhibit reactivity towards these proteins. A concentration of 0.5 µg/mL, within the reported range for NS1 in the serum of DENV-infected patients, was used [18,19]. Additionally, the detection of NS1 in pools of DENV-positive and -negative sera was explored to evaluate the clinical applicability of the systems. Both systems successfully detected recombinant NS1 proteins from DENV serotypes 1 to 4, with varying absorbance values (Fig 3). High sensitivity was observed for NS1-1 and NS1-2, while detection of NS1-3 and NS1-4 was less efficient. No cross-reactivity with ZIKV NS1 was detected, confirming the specificity of both systems for distinguishing DENV infections from ZIKV. However, neither system was able to detect the NS1 protein present in patient sera.

The failure to detect NS1 in positive serum samples may be attributable to immune complex formation between NS1 and anti-NS1 antibodies. This interaction potentially masks NS1 epitopes, thereby preventing recognition by detection Nbs. Various treatments, including heat and pH modifications using acidic and alkaline buffers, were employed to dissociate these immune complexes and destabilize antigen-antibody interactions [65–67]. Despite these efforts, no improvement in NS1 detection was observed in the treated samples. Nanobodies from the initial panel, assessed in both their monovalent and Fc-fused formats, were evaluated for their ability to capture NS1 in clinical serum samples. Under the assay conditions employed, none of the candidates exhibited detectable NS1 capture (S11 Fig).

## Development of a combined Nanobody ELISA for dengue diagnosis

Polyclonal anti-NS1 serum was produced in a rabbit following a five-dose immunization protocol. Indirect ELISA results demonstrated strong binding of the purified IgG to recombinant NS1 proteins from all four DENV serotypes, with absorbance values increasing in a dose-dependent manner (S12 Fig). Endpoint titration showed that a minimum IgG concentration of 0.2 µg/mL was sufficient to detect

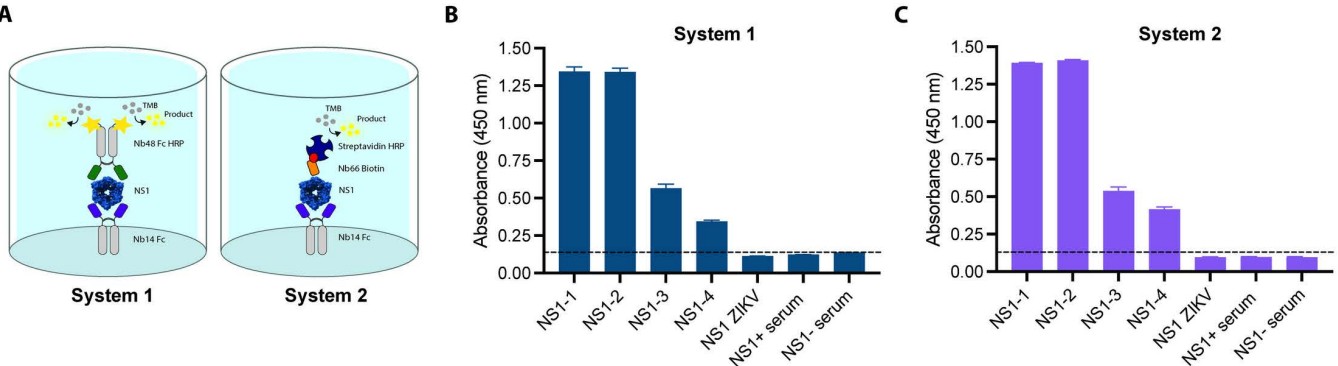

**Fig 3. Detection of recombinant DENV and ZIKV NS1 proteins in serum matrix, and pools of DENV-positive and -negative sera using the proposed systems.** A-Schematic representation of the two proposed Nanobody-based ELISA systems. System 1: 4 µg/mL of Nb14 Fc as the capture antibody and 10 µg/mL Nb48 HRP as the detection antibody. System 2: 2 µg/mL of Nb14 Fc as the capture antibody and 5 µg/mL biotinylated Nb66 as the detection antibody. B-Reactivity of System 1 with NS1 variants from DENV serotypes 1–4, ZIKV NS1, and human sera. C-Reactivity of System 2 with the same antigen panel. Data is shown as mean absorbance at 450 nm ± SD (n=3). Black dashed line represents the mean OD of negative control samples plus 3 standard deviations.

NS1 from all DENV serotypes. Mild cross-reactivity with ZIKV NS1 was observed, likely due to the polyclonal nature of the serum, but this was limited to high antibody concentrations and was significantly weaker than binding to DENV NS1 (S12 Fig).

The purified polyclonal IgG, combined with biotinylated Nb66 as a detection antibody, was employed to develop a cNb-ELISA for DENV NS1 (Fig 4A ). Nanobody 66 was selected for its cost-effectiveness, reduced usage concentration, and strong performance in Nb-based ELISA. The ability of purified IgG to capture recombinant NS1 from DENV serotypes 1–4 and ZIKV was assessed in a serum matrix devoid of anti-NS1 reactivity. At an antigen concentration of 0.5 µg/mL, the IgG successfully captured NS1-1 and NS1-2 with high sensitivity, while NS1-3 and NS1-4 exhibited lower absorbance values (Fig 4B). No cross-reactivity with ZIKV NS1 was detected when the purified IgG was used as the capture antibody, confirming the system's specificity for distinguishing DENV infections from ZIKV. To further evaluate the specificity of our detection system for dengue virus NS1 protein, we tested supernatants from Vero cells infected with DENV serotypes 1–4, ZIKV, CHIKV, and YFV. The selected Nbs showed no cross-reactivity with other flaviviruses, including YFV and Zika virus, nor with the alphavirus CHIKV, which presents clinical symptoms that closely resemble those of dengue (Fig 4D).

The optimal concentrations of capture and detection antibodies for the cNb-ELISA were determined using a previously described chessboard titration method. Based on these findings, 10 µg/mL of purified polyclonal anti-NS1 serum and 5 µg/mL of biotinylated Nb66 were selected for the assay. In contrast to the previously developed Nb-only systems, the cNb-ELISA effectively detected NS1 in a pool of positive serum samples (Fig 4C). At both tested dilutions (1:5 and 1:10), the assay produced a clear and distinguishable signal for positive samples compared to negatives, underscoring its robustness and sensitivity. Its demonstrated ability to detect NS1 in clinical samples highlights the potential of this cNb-ELISA as a diagnostic tool, successfully addressing the limitations of earlier systems.

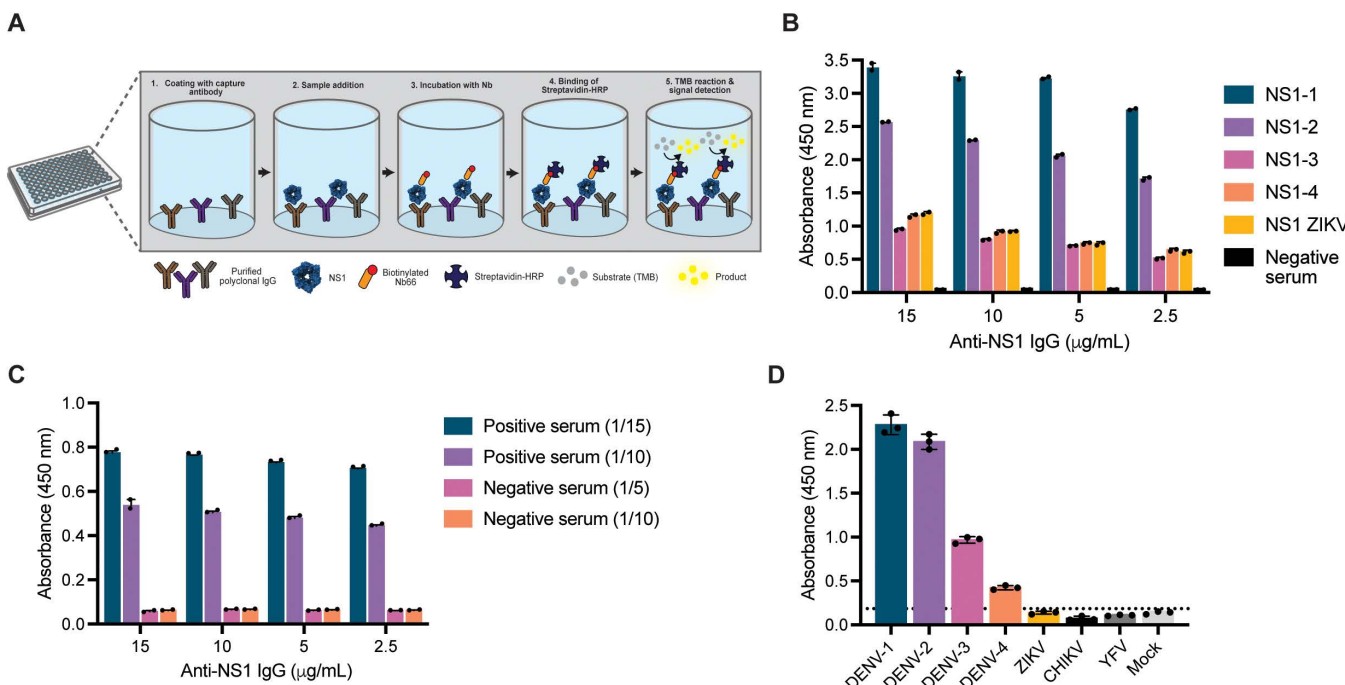

**Fig 4. Evaluation of the combined Nanobody ELISA for the detection of DENV NS1.** A-Schematic representation of the combined Nanobody ELISA workflow. B-Detection of recombinant NS1 proteins from DENV and ZIKV in a negative serum matrix. C-Detection of NS1 in pooled human serum samples from DENV-positive and negative patients. Serum dilutions (1:5 and 1:10) were analyzed using biotinylated Nb66 as detector. D-Specificity assessment of the combined Nb ELISA using supernatants from Vero cells infected with different flaviviruses (DENV-1 to DENV-4, ZIKV, YFV) and the unrelated arbovirus CHIKV, which co-circulate in endemic regions and cause overlapping clinical symptoms. Mock-infected supernatant was used as a negative control.

## Validation of the combined Nanobody ELISA

To evaluate the performance of the cNb-ELISA, serum samples were collected from individuals presenting symptoms consistent with DENV infection. A flow diagram of the validation process for the cNb-ELISA is shown as supplementary information (S13 Fig). These samples were categorized at Orán Hospital as either NS1-positive (n=94, confirmed using PLATELIA DENGUE NS1 Ag from BioRad) or epidemiologically negative (n=74, not tested with by a commercial kit due to being collected 5 or more days after the acute phase). The ROC curve analysis yielded an area under the curve (AUC) of 0.9408 with a 95% confidence interval (CI) of 0.9078 to 0.9730. This indicates excellent discriminative capacity (AUC > 0.9; Fig 5A). Using the Youden index and the MaxSpSe method, a cutoff absorbance value of 0.126 at 450 nm was established. At this threshold, the diagnostic sensitivity was 93.6% (95% CI: 86.6–97.6%) and the diagnostic specificity was 81.1% (95% CI: 70.3–89.3%). Samples with absorbance values below 0.126 were classified as negative, while those exceeding this threshold were classified as positive (Fig 5B).

Agreement of the developed cNb-ELISA with results obtained after the application of the official algorithm executed at Orán Hospital (NS1-positive samples analyzed using the Bio-Rad test for specimens collected up to day five, along with epidemiologically negative samples) was calculated. Kappa calculated (K=0.75) using data from Table 3 indicates a substantial agreement according to the standard interpretation by Landis and Koch [68].

Regarding discordant results (S2 Table), 6 samples initially identified as positive were negative by the cNb-NS1 ELISA, while 14 samples identified as negative tested positive. To address this discrepancy, the samples were re-evaluated using the commercial ELISA Dengue virus NS1 antigen (DIA.PRO). Six samples undetected by our cNb-ELISA were confirmed as positive by the DIA.PRO results, highlighting a limitation in our assay. However, five positive samples were negative both for the cNb-ELISA and the DIA.PRO ELISA. It is likely that these positive results were inaccurate, as the unusually low cutoff reported for these samples using the Bio-Rad test stands out when compared to the cutoffs of the other samples measured with the same kit (S3 Table). Therefore, these group of samples were not included in this analysis.

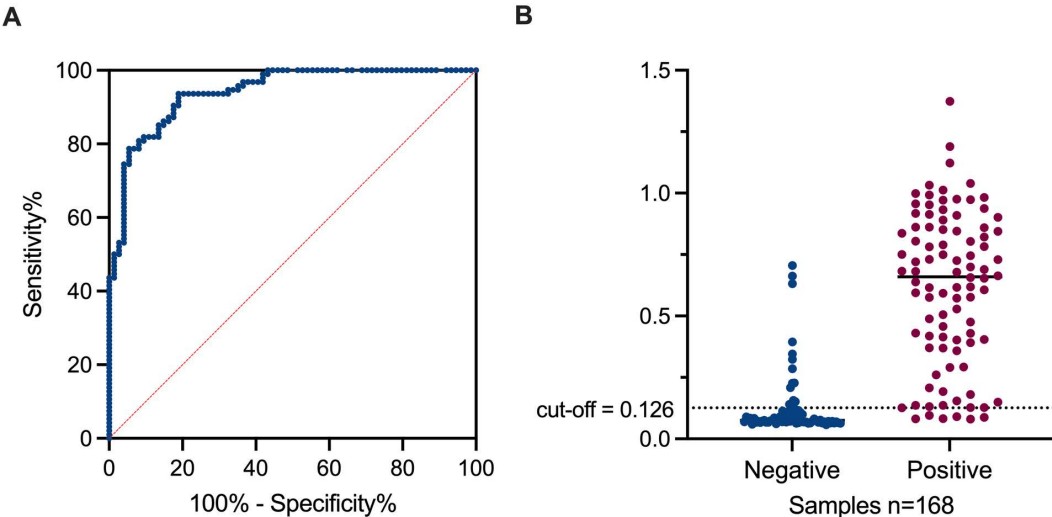

**Fig 5. Performance of the developed cNb-ELISA.** A-Receiver operating characteristic (ROC) curve analysis indicating the calculated AUC for the validation of the assay. B-Interactive dot plot showing the absorbance value of each serum sample. Dot line shows the determined cut-off point.

**Table 3. Concordance Analysis between the cNB-ELISA test and the results from the algorithm executed at Orán Hospital.**

| | | Reference samples | | |
|---|---|---|---|---|
| | | positive | negative | total |
| **cNb-ELISA** | positive | 88 | 14 | 102 |
| | negative | 6 | 60 | 66 |
| | total | 94 | 74 | |

Orán Hospital adheres to an "algorithm for the differential diagnosis of febrile syndrome in its initial phase", established by the health authority, to analyze human samples. According to this algorithm, samples collected five days after symptom onset are not tested but can be classified as epidemiologically positive or negative. Fourteen samples epidemiologically negative from this group tested positive with our kit, and the results were corroborated by the DIA.PRO test (S2 and S3 Tables). This highlights the critical role of antigen detection in diagnosing dengue infection while emphasizing the effectiveness of our system in identifying NS1 in samples collected over five days after symptom onset.

## Analytical sensitivity and limit of detection of the combined Nanobody ELISA

The analytical sensitivity of the cNb-ELISA was evaluated using recombinant NS1 protein spiked into human serum (Fig 6). Standard curves were performed for each serotype by using NS1 at different concentrations and data was fitted to a 4-parameter logistic model. The LOD values were determined to be 101.80 ng/mL, 133.68 ng/mL, 507.70 ng/mL, and 425.58 ng/mL for NS1-1, NS1-2, NS1-3, and NS1-4, respectively (Fig 6A).

Absorbance values above the cutoff point were obtained up to a 1/256 dilution of the pool of positive sera form infected patients, confirming the high sensitivity of the system. In contrast, the negative sera used as controls did not show absorbance values above the proposed cutoff point at any of the dilutions evaluated. From this assay, the linear range of the ELISA was determined, defined by the dilutions of the sera pool in which the absorbance is proportional to the antigen concentration. Within this range, a significant linear relationship was obtained, with a determination coefficient ($R^2$) of 0.997, indicating an excellent correlation between the sera pool dilutions and the detected signal (Fig 6B). These results

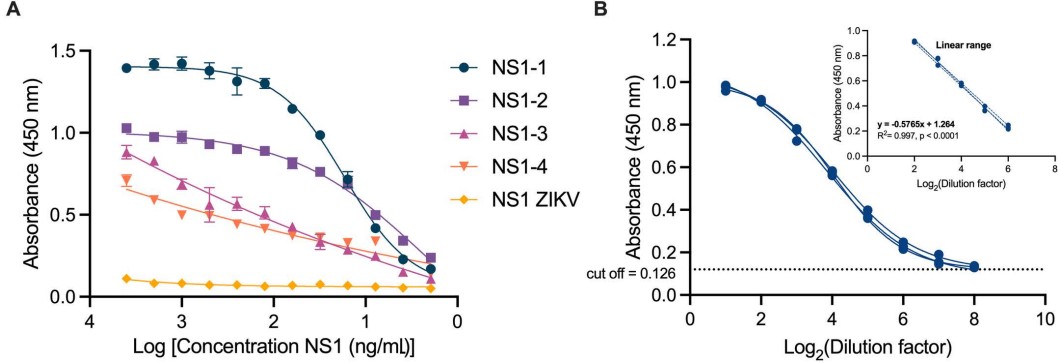

**Fig 6. Determination of the limit of detection and the analytical sensibility.** A-ELISA to detect DENV1-4 NS1 at different concentrations to determine the limit of detection of the optimized cNb-ELISA. The developed detection system has variable LOD for the different DENV NS1. B-Estimation of the analytical sensitivity of the cNb-ELISA, absorbance values obtained based on 2-fold serial dilutions of the pool of positive sera in a range from 1/2 to 1/256. Inset: linear regression analysis of absorbance values based on antigen concentration.

demonstrate the robustness of the developed system, as it can maintain high sensitivity and reproducibility over a wide range of dilutions of the positive sera pool.

## Discussion

Dengue in Argentina has evolved from sporadic outbreaks into a persistent public health challenge [8]. Alarmingly, only 35% of cases in 2024 were confirmed through laboratory testing, underscoring a substantial diagnostic gap and widespread underreporting [8]. These findings emphasize the pressing need to strengthen diagnostic infrastructure and develop cost-effective tools, especially considering the high demand and expense of imported commercial kits.

NS1 is a multifaceted and complex viral protein, with its secreted form recognized as an exceptional disease biomarker [17]. Importantly, the structure of NS1 comprises several similar components across different flaviviruses, which could result in the cross-reactivity of antibodies or detection tools [12]. Previous studies have demonstrated that high serum levels of secreted NS1 can be detected as early as the first day after the onset of fever and may correlate with disease severity [12,17–19]. Several diagnostic methods are available to detect this biomarker [33–38]. However, the NS1 ELISA diagnostic test stands out due to its numerous advantages, including early detection, high sensitivity and specificity, cost-effectiveness, and ease of use.

Although there are several commercial DENV NS1 ELISA kits available, in Argentina, but also in Latin-American countries, most public hospitals use Platelia Dengue NS1 Ag kit (BioRad) and Panbio Dengue Early ELISA kit (Abbott). These kits are imported and impose a significant financial burden on the public health system. In recent years, their acquisition has been further hindered by customs restrictions. This underscores the urgent need for a locally developed detection system. Recently, the ELISA test Detect-AR Dengue (Laboratorio Lemos) was approved by the National Medicines, Food, and Medical Technology Authority (ANMAT) under Regulation No. 1-0047-3110-006321-24-9. This monoclonal antibody-based detection system has proven effective in identifying dengue NS1. However, it is important to consider that the production costs of monoclonal antibodies, which rely on eukaryotic cell lines and expensive culture media, are significantly higher than those of Nbs produced in bacteria [69,70].

Nanobodies have been successfully used to inhibit viral infections and detect various pathogens [43,44]. A Nb that specifically recognizes the DENV-2 NS1 performed better than a monoclonal antibody in immunochromatographic test strips [71]. Also, an optimized MagPlex assay, based on the use of Nbs, allowed the detection of four DENV serotypes on spiked serum samples, and no cross-reactivity to other flavivirus was observed [71]. Even more, the detection of NS1 was increased by a factor of 5 when using oriented dimers using the SpyTag/SpyCatcher system for capture [72]. A common limitation in prior studies is the use of recombinant proteins, spiked serum, or a limited number of infected human samples—insufficient for robust diagnostic validation.

In this study, a llama was immunized with NS1 protein derived from insect cells (C6/36). Multiple studies have demonstrated that NS1 glycoproteins from insect and mammalian hosts are secreted and exist in various oligomeric forms [12,14,73]. Furthermore, NS1 has been reliably detected in the supernatant of C6/36 cells infected with different DENV serotypes using commercial ELISA kits [74–76], suggesting the presence of conserved epitopes. Secretion kinetics vary across cell types and flaviviruses, in the case of DENV, Alcalá *et al.* demonstrated that C6/36HT cells begin secreting soluble NS1 from DENV-2 and DENV-4 as early as 6 hours post-infection [77]. Ludert *et al.* observed similar secretion kinetics in C6/36HT cells, albeit with a 12-hour delay compared to Vero cells [74]. Or results confirm that NS1 was secreted into the supernatant following infection of C6/36 cells with DENV as immunization with this material successfully elicited a specific immune response, supporting its suitability for antibody generation despite the host origin.

After employing two distinct selection strategies, we successfully identified several Nbs exhibiting significant variability in their CDR3 sequences, which were categorized into different families (S4 and S5 Figs). These Nbs demonstrated the

ability to specifically detect the NS1 protein across all four DENV serotypes, with varying levels of relative affinity depending on the viral serotype (S6 Fig). Notably, none of the Nbs showed recognition of the Zika NS1 protein.

Our results showed that multimerizing Nbs, either through Fc fusion or bivalent design, significantly enhanced their ability to capture recombinant NS1 protein. However, despite the increased affinity of bivalent constructs, their capture performance did not exceed that of Fc-fused Nbs (Fig 2). Previous studies have consistently demonstrated that Fc fusion improves binding, neutralization, and overall functional activity [78,79]. Notably, Anbuhl *et al.* reported superior binding and receptor clustering with an Fc-coupled CXCR4-targeting Nb compared to its bivalent counterpart. This advantage was attributed to the structural rigidity of Fc fusion at the C-terminus, in contrast to the flexible GS-linker used in bivalent formats [80]. Although bivalent Nbs can engage both paratopes, their apparent affinity may be compromised by linker flexibility, increased molecular size, or altered kinetics. Importantly, bivalent constructs remain valuable for low-cost diagnostic platforms, as they can be efficiently produced in bacterial systems—offering a more economical alternative to Fc-fused Nbs, which require eukaryotic expression systems.

To reduce production costs of detection Nbs, two Nb biotinylation methods were tested. The AviTag system enables rapid biotinylation through the interaction of periplasmic Nbs with cytoplasmic BirA in the presence of biotin [55]. While the biotinylation was effective, the system showed very low efficiency (S10A Fig) despite its ability to use biotin from various sources, including low-cost options like pharmacy-grade hair care products containing biotin (S10B Fig). To address this, we adopted a more efficient system that achieves 100% biotinylation by incorporating a purification step to remove His-tagged non-biotinylated Nbs [56]. Additionally, transiently transfected HEK-293T cell lines have been successfully used to express Nb-HRP fusions [81,82]. Establishing a stable CHO cell line for this purpose could further reduce production costs and streamline the generation of Nb-based detection reagents.

After identifying the optimal capture antibody (Fig 1B), various dilutions of this Nb, along with potential detection Nbs, were tested (S8 and S9 Figs). Following the determination of the most effective antibody pair, efforts were focused on establishing optimal conditions for NS1 detection in patient samples. Unfortunately, our Nbs were unable to capture the NS1 protein from the serum of infected patients. However, they successfully recognized low concentrations of recombinant protein or spiked serum (Fig 3). A possible explanation for this limitation is that the recombinant protein, whether purified or derived from cells transfected with the same construct used for Nb selection, may have adopted a structure that failed to achieve proper conformation, or one dissimilar to the native NS1 protein present in patient sera. NS1 is a highly complex protein, with multiple reported structures that vary depending on its interaction with different components and the glycosylation capacity of the cell line in which it is produced [73,83,84]. On the other hand, a recent study demonstrated that Nbs produced against inactivated botulinum neurotoxin recognized conformational epitopes that were significantly altered when the toxin was directly immobilized on the plastic surface of the ELISA plate [85]. This suggests that employing a capture strategy for the protein, against which Nbs are to be selected, would be a more effective approach than direct immobilization on the plate. A more effective approach might involve using a selection strategy directly with sera from positive patients, while counter-selecting with negative sera, rather than relying on recombinant protein. We are currently adopting this new strategy, as it seems promising for selecting Nbs with strong capture properties from the existing library. This is because the llama was immunized with protein purified from the supernatant of C6/36 insect cells, which are known to produce the soluble hexameric form of NS1 [86]. The primary motivation for this approach is that developing a detection system for DENV-infected patients relying solely on Nbs would significantly lower production costs and enhance the reproducibility of detection, a known challenge when using polyclonal sera.

It has been reported that monovalent Nbs exhibit limited performance as capture antibodies when directly immobilized on polystyrene microplates, and that increasing Nbs binding sites could improve their performance on an ELISA [87–89]. Different strategies to multimerize Nbs and to improve antigen capture have been applied such as multimerization through ferritin [90,91], binding to iodoacetyl-functionalized pullulan [92], coupling to streptavidin [93] or GFP [94], among other. Considering these findings, it is also possible that the use of bivalent or Fc-coupled Nbs may not suffice to capture a

complex antigen such as the NS1 protein. This limitation could potentially be overcome by employing Nbs cocktails, however this strategy did not work in our hands.

In this study, to address the capture limitations, we immunized a rabbit with DNA vaccines and recombinant NS1 proteins. We observed that an IgG concentration as low as 0.2 µg/mL was sufficient to detect NS1 from all DENV serotypes, with only mild cross-reactivity observed with ZIKV NS1 (S12 Fig). Li *et al.* employed similar strategies, testing various ELISA formats using polyclonal antibodies and Nbs [95]. Their findings revealed that combining polyclonal sera as the capture antibody with Nbs —either biotinylated or HRP-conjugated—enabled highly sensitive and selective detection of mouse soluble epoxide hydrolase protein in a sandwich immunoassay.

The limit of detection of the cNb-ELISA varied significantly across DENV serotypes, ranging from 101.8 to 507 ng/mL. These LODs were higher than those reported for immunochromatographic assays, but the latter were measured using a protein-free serum substitute enriched only with BSA, which does not reflect the complexity of human sera [30]. Similarly, lower LODs were reported for DENV-2 using the same approach, though this evaluation was conducted in 0.85% NaCl, without considering the intricacies of a serum matrix [96]. Likewise, the LOD determined using a MagPlex assay was in the low nanogram range for all DENV serotypes [71].

Using a total of 94 positive and 74 negative clinical samples provided by the Orán Hospital, as reference sample population, we were able to validate the cNb-ELISA. Selecting a cut off =0.126, the assay showed a diagnostic sensitivity of ~93.6% and a diagnostic specificity of ~81.1%. The sensitivity of the cNb-ELISA surpasses that of the Platelia Dengue NS1 Ag kit, which is reported to have a sensitivity ranging from 60% to 88% [97–99], and that of the Panbio Dengue Early ELISA kit shows sensitivity ranging from 56.4% to 96.0%, depending on the DENV serotype [100,101]. However, the specificity of our detection system is lower than that reported for both commercial kits, which have specificities ranging from 97.4% to 100% [100]. The sensitivity and specificity of the cNb-ELISA could have been higher if negative samples had been tested using the same Bio-Rad kit as the positive ones. If the 14 samples epidemiologically classified as negative were instead considered positive—given that both our kit and the DIA.PRO kit yielded positive results (S2 and S3 Table)— these values would have been greater and would more accurately reflect the performance of our detection system. In the same sense, the concordance analysis of the developed cNb-ELISA with results obtained after the application of the official algorithm executed at Orán Hospital, yielded a Kappa value of 0.75. This represents a substantial agreement, as defined by the standard interpretation of Landis and Koch [68]. This value could have also been higher if the samples initially classified as epidemiologically negative had been reclassified as positive, as both our detection system and the DIA. PRO kit produced consistent results (S2 and S3 Tables). Unfortunately, we were unable to test all samples using the DIA. PRO kit due to the limited availability of sera.

A deeper analysis of the discordant samples reveals certain limitations in the ability of the cNB-ELISA to detect some positive cases. Nevertheless, it demonstrates a distinct advantage by successfully detecting NS1 beyond the fifth day after symptom onset. The inability to detect certain positive samples from 2016 could be linked to the cNb-ELISA's limited capacity to identify the NS1-4 protein at low concentrations (Fig 6A). During this period, cases of DENV-4 were reported in Salta Province (General Güemes, Orán, and Aguaray), where the samples were collected [10]. Consequently, detection failure may be associated with the specific viral serotype infecting these patients. Notably, lower sensitivity for DENV-4, as well as false negative results for this serotype have been reported using commercial kits [101,102]. It is important to note that DENV-4 NS1 is phylogenetically more distant from other DENV NS1 variants [103]. Furthermore, studies have reported that human antibodies capable of detecting NS1-1, NS1-2 and NS1-3 fail to recognize NS1-4 [104]. NS1 antigen sensitivity has been shown to vary across different epidemiological contexts. Felix *et al.* reported that secondary DENV-2 infections may have contributed to a high rate of false-negative results during the 2010 dengue outbreak in Brazil [105]. Sequencing of NS1-positive and -negative isolates revealed no mutations to explain the diagnostic failure. Koraka *et al.* later suggested that reduced NS1 detection in secondary infections may result from the sequestration of NS1 protein within immune complexes, limiting its availability for capture [106]. In our case, immune complex dissociation treatments were applied only to

the pooled positive serum samples using Nbs as the capture element. We were unable to perform this treatment on individual samples in the cNb-ELISA due to the limited volume of samples available for testing. On the other hand, the cNb-ELISA was able to detect 14 positive samples (later confirmed by the DIA.PRO kit, S2 and S3 Tables) that were considered epidemiologically negative. These samples were not tested by commercial kits due to the application of the algorithm for febrile syndrome established by the health authority as they were taken 5 days after the onset of symptoms. These results highlight the need to re-consider the algorithm used for the monitoring of dengue infection in public hospitals.

Rotadial represents the first Nb-based immunoassay approved for diagnostic use in Argentina, specifically developed to detect Group A Rotavirus in pediatric fecal samples. Utilizing a sandwich ELISA format with Nbs targeting the VP6 protein, the assay demonstrated 100% sensitivity and 99% specificity when benchmarked against commercial kits [107]. Rotadial has been incorporated into Argentina's national diagnostic network, with thousands of units distributed to hospitals participating in the Red Nacional de Vigilancia de Gastroenteritis Virales [108]. Its implementation underscores the practical viability, affordability, and diagnostic robustness of Nb-based platforms in clinical settings.

In summary, our study highlights the urgent need for accessible and affordable dengue diagnostic tools, particularly in endemic regions like Argentina, where imported kits are limited by cost and availability. We successfully generated and characterized Nbs capable of detecting NS1 from all four DENV serotypes with no cross-reactivity to ZIKV, CHKV or YFV (Fig 4D). While bivalent and Fc-tagged constructs improved antigen capture in recombinant systems, limitations in detecting native NS1 in patient sera emphasize the complexity of developing a robust detection system. The cNb-ELISA demonstrated high diagnostic sensitivity and promising clinical performance, including the detection of cases beyond the fifth day of symptoms—an important advantage over existing executed algorithms. However, further optimization is needed to improve specificity and ensure reliable detection across all serotypes, particularly DENV-4. These findings reinforce the potential of Nbs-based assays as a sustainable alternative for dengue diagnosis and highlight the need to revise current diagnostic algorithms to improve detection of late-presenting cases.

## Supporting information

**S1 Table. List of primers used in this study.**
(XLSX)

**S2 Table. Discordant results between Platelia Dengue NS1 Ag kit (BioRad) from Orán Hospital, DIA.PRO kit and cNb-ELISA.**
(XLSX)

**S3 Table. Sample OD, cutoff and result interpretation for every sample used in this study according to the Platelia Dengue NS1 Ag kit (BioRad) from Orán Hospital, the developed cNb-ELISA or DIA.PRO kit.**
(XLSX)

**S1 Fig. Llama antibody titers and efficiency of the Nanobody library. A**-IgG antibody titers were evaluated before and after immunization with purified supernatants containing the NS1 protein from each DENV strain. Higher titers were achieved for NS1-2 (dilution 1/1350), NS1-3 (dilution 1/4050) and NS1-4 (dilution 1/1350). In contrast, the anti-NS1-1 titer showed no significant increase even after multiple immunizations (dilution 1/450). **B**-Agarose gel displaying ~700 bp PCR fragments corresponding to VHHs of varying sizes, and ~300 bp PCR fragments corresponding to empty vectors amplified from randomly selected individual colonies. The efficiency of the library was determined to be 78.5% (33 out of 42 colonies contain fragments incorporating a VHH sequence).
(JPG)

**S2 Fig. Determination of NS1 cross-reactivity of the selected Nanobodies by ELISA.** In the upper panel, clones selected using NS1-1 were tested against NS1-1, NS1-2, non-coated wells (PBS or supernatant of non-transfected cells),

and an irrelevant His-tagged protein. In the lower panel, clones selected with NS1-2 were tested against NS1-2, NS1-1, non-coated wells (PBS), and an irrelevant His-tagged protein. Several cross-reactive clones were identified.
(TIF)

**S3 Fig. Restriction Fragment Length Polymorphism analysis of selected clones.** RFLP analysis showing patterns obtained after restriction digestion of PCR fragments with HinfI, a high frequency cutting restriction enzyme. Samples were analyzed on a 2.5% agarose gel and stained with ethidium bromide. At least 18 distinct patterns were identified (arrows).
(JPG)

**S4 Fig. Logo plot of unique Nanobody sequences.** A sequence logo plot was generated using 11 unique Nb sequences with WebLog3, highlighting high variability in the CDR3 domains.
(JPG)

**S5 Fig. Phylogenetic and bioinformatic analysis of the selected Nanobodies.** Phylogenetic analysis of Nbs selected against NS1-1 and NS1-2 identified three main clusters (red, orange, and blue). Group 1 (Nb5, Nb34, Nb51, and Nb66) exhibited an 18-amino-acid CDR3 and consistent V gene 3-301 usage, along with J genes 401 or 201/301. Group 2 (Nb48 and Nb54) featured a longer CDR3 (21 amino acids) and displayed variable J gene usage, as determined by DomainGapAlign. Group 3 (Nb7 and Nb22) had a compact 10-amino-acid CDR3 with consistent V gene 3S5301 and J gene 401 usage. Nanobodies Nb14, Nb40, and Nb70 did not cluster within these groups and instead displayed unique genetic and structural characteristics, with Nb40 having the longest CDR3 sequence among all analyzed variants. Germline origins were determined using IMGT/V-QUEST (columns 3–5) and IMGT/DomainGapAlign (columns 6–7), while CDR3 length and biopanning strategies are detailed in columns 8 and 9.
(JPG)

**S6 Fig. Evaluation of the binding capacity of Nanobodies to NS1 from DENV and ZIKV.** ELISA plates were coated with recombinant NS1 proteins and purified HRP-coupled Nbs serially diluted were added. The absorbance curves at 450 nm show the specific binding profile of each Nb to the different DENV NS1. None of the Nbs showed reactivity toward ZIKV NS1.
(TIFF)

**S7 Fig. Identification of shared epitopes recognized by DENV NS1-specific Nanobodies** A competitive binding assay was conducted using Fc-fused Nbs and their monovalent counterparts to assess epitope overlap. The results indicate that Nb48 and Nb14 compete for the same binding site on the NS1 protein, suggesting recognition of a shared epitope. In contrast, Nb66 does not compete with either Nb48 or Nb14, indicating that it targets a distinct epitope.
(TIFF)

**S8 Fig. Optimization of the Nb-ELISA using HRP conjugated Nanobodies.** Absorbance values at 450 nm obtained from two-dimensional titration using different concentrations of Nb14 Fc as the capture antibody and Nb48 HRP (A) or Nb66 HRP (C) as the detection antibody. The assay was performed with and without 0.5 µg/mL of NS1-2. P/N ratio for combinations of Nb14 Fc and Nb48 HRP (B) or Nb66 HRP (D), indicating the best combination for the highest P/N ratio. The P/N ratio is calculated by dividing the absorbance of positive samples (NS1-2) by the absorbance of negative samples (without NS1-2).
(TIFF)

**S9 Fig. Optimization of the Nb-ELISA using biotinylated Nanobodies.** Absorbance values at 450 nm obtained from two-dimensional titration using different concentrations of Nb14 Fc as capture antibody and biotinylated Nb48 (A) or biotinylated Nb66 (C) as the detection antibody. The assay was performed with and without 0.5 µg/mL of NS1-2. P/N ratio

for combinations of Nb14 Fc and biotinylated Nb48 (B) or biotinylated Nb66 (D), indicating the best combination for the highest P/N ratio. The P/N ratio is calculated by dividing the absorbance of positive samples (NS1-2) by the absorbance of negative samples (without NS1-2).
(TIFF)

**S10 Fig. Comparison of biotinylation systems.** Nanobodies were biotinylated using either the AviTag system or the sortase A system. **A-**The left panel shows a western blot probed with an anti-His-HRP antibody, detecting His-tagged Nbs biotinylated via the AviTag system, which retains the His tag during biotinylation. No signal was observed for Nbs biotinylated with the sortase A system, as the His tag is removed during the reaction. The right panel presents a western blot of the same samples detected with Streptavidin-HRP, showing a weak signal for Nbs biotinylated via the AviTag system and a stronger signal for those biotinylated using the sortase A system. **B-**Western blot using Streptavidin-HRP to detect Nbs biotinylated via the AviTag system using A: commercial D-(+)-Biotin (catalogue 2031, Sigma-Aldrich) or B: Fast Dissolve Biotin 10.000 mcg, vitamin supplement (Carlyle).
(TIF)

**S11 Fig. Evaluation of monovalent and Fc-fused Nanobodies for NS1 detection in clinical serum samples.** The ability of Nbs, in both monovalent and Fc-fused formats, to capture NS1 from clinical serum samples was assessed using the established cNb-ELISA protocol. None of the tested Nbs demonstrated detectable NS1 capture under the conditions employed.
(TIFF)

**S12 Fig. Titration curves of purified IgG from rabbit anti-NS1 polyclonal serum.** Purified rabbit IgG obtained after a five-dose immunization protocol was serially diluted to detect NS1 proteins (2 µg/mL). Specific recognition was detected using an anti-rabbit HRP-coupled antibody. Each experiment was conducted in duplicate.
(TIF)

**S13 Fig. Flow diagram of the validation process for the combined Nanobody ELISA.** Human serum samples were collected at Orán Hospital, Salta (2016–2019) and classified by a reference diagnostic test (positive, n=94; negative, n=74). Samples were analyzed using the cNb-ELISA, and results were subjected to statistical validation. Diagnostic validation included ROC curve analysis (sensitivity, specificity, AUC, and cut-off determination), followed by assessment of diagnostic performance (sensitivity, specificity, Cohen's kappa agreement, and DIA-PRO analysis for discordant samples). Analytical validation included determination of sensitivity and limit of detection (LOD).
(TIF)

## Acknowledgments

We thank Dr. Alicia Mistchenko, Hospital de Ninos Dr. Ricardo Gutierrez, Buenos Aires, Argentina for sharing the virus strain used in this work. We thank José Manuel Gomez for technical assistance. We acknowledge Mr. Ismael Anze for all his help with the procedures performed to gain access to patient samples. We thank David Liu for providing us with sortase A pentamutant (Addgene plasmid #75144) and to Mitchell O'Connell for sharing the dual vector Avitag-BirA (Addgene plasmid # 100817). The following reagent was obtained through BEI Resources, NIAID, NIH: Monoclonal Anti-Dengue Virus Type 1 Nonstructural Protein 1, Clone 15F3-1 (produced in vitro), NR-10121.

## Author contributions

**Conceptualization:** María Florencia Pavan, Lorena Itatí Ibañez.

**Data curation:** María Florencia Pavan.

**Formal analysis:** María Florencia Pavan, Cristian Miguel Malnero, María Emilia Mora Alvarado, Viviana Parreño, Lorena Itatí Ibañez.

**Funding acquisition:** Lorena Itatí Ibañez.

**Investigation:** María Florencia Pavan, Cristian Miguel Malnero, María Emilia Mora Alvarado, María Camila Carzoglio, Yesica Paredes Rojas, Laura Perez Vidakovics, Cybele Carina García, Lorena Itatí Ibañez.

**Methodology:** María Florencia Pavan, Cristian Miguel Malnero, María Emilia Mora Alvarado, Leo Hanke, Alejandro Castello, Lorena Itatí Ibañez.

**Project administration:** Lorena Itatí Ibañez.

**Resources:** Agostina Bruno, Alejandro Castello, Gerald McInerney, Lorena Itatí Ibañez.

**Supervision:** Lorena Itatí Ibañez.

**Validation:** María Florencia Pavan, Viviana Parreño.

**Visualization:** María Florencia Pavan, Lorena Itatí Ibañez.

**Writing – original draft:** María Florencia Pavan, Viviana Parreño, Lorena Itatí Ibañez.

**Writing – review & editing:** María Florencia Pavan, Cristian Miguel Malnero, Laura Perez Vidakovics, Leo Hanke, Alejandro Castello, Gerald McInerney, Cybele Carina García, Viviana Parreño, Lorena Itatí Ibañez.

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
