## [Decision Letter · Decision Letter 0]

8 Jul 2025

Engineered Nanobodies for early and accurate diagnosis of dengue virus infection

Dear Dr. Ibañez,

Thank you for submitting your manuscript to PLOS Neglected Tropical Diseases. After careful consideration, we feel that it has merit but does not fully meet PLOS Neglected Tropical Diseases's publication criteria as it currently stands. Therefore, we invite you to submit a revised version of the manuscript that addresses the points raised during the review process.

Please submit your revised manuscript within 60 days Sep 06 2025 11:59PM. If you will need more time than this to complete your revisions, please reply to this message or contact the journal office at plosntds@plos.org. Please include the following items when submitting your revised manuscript:

We look forward to receiving your revised manuscript.

Kind regards,

Guy Caljon

Academic Editor

David Safronetz

Section Editor

Shaden Kamhawi

co-Editor-in-Chief

Paul Brindley

co-Editor-in-Chief

**Additional Editor Comments :**

Your manuscript was reviewed by 3 independent reviewers and an editorial board member. While the reviewers appreciate the presented work, a major revision will be needed for the manuscript to be considered for publication in PLoS NTDs. Several issues were raised about the use of a polyclonal serum to generate a capture reagent, epitope specificity of the Nbs, the analysis of diagnostic accuracy, and cross-reactivity with other flaviviruses. In addition, the reviewers provided comments aimed at improving the manuscript’s background on NS1-based diagnostics, enhancing methodological detail for reproducibility, and reducing redundancy in the discussion section.

**Journal Requirements:**

2) We noticed that you used the phrase 'data not shown' in the manuscript. We do not allow these references, as the PLOS data access policy requires that all data be either published with the manuscript or made available in a publicly accessible database. Please amend the supplementary material to include the referenced data or remove the references.

- TM on pages: 11, and 19.

4) Thank you for including an Ethics Statement for your study. Please include:

i) A statement that formal consent was obtained (must state whether verbal/written) OR the reason consent was not obtained (e.g. anonymity). NOTE: If child participants, the statement must declare that formal consent was obtained from the parent/guardian.].

5) We have noticed that you have uploaded Supporting Information files, but you have not included a complete list of legends. Please add a full list of legends for your Supporting Information files (Supplementary Tables) after the references list.

**Reviewers' Comments:**

Reviewer's Responses to Questions

**Key Review Criteria Required for Acceptance?**

**Methods**

-Are the objectives of the study clearly articulated with a clear testable hypothesis stated?

-Is the study design appropriate to address the stated objectives?

-Is the population clearly described and appropriate for the hypothesis being tested?

-Is the sample size sufficient to ensure adequate power to address the hypothesis being tested?

-Were correct statistical analysis used to support conclusions?

-Are there concerns about ethical or regulatory requirements being met?

Reviewer #1: 1. The methodology primarily focused on optimizing the ELISA conditions (e.g., antibody concentrations and formats) through chessboard titration experiments, rather than employing detailed epitope mapping. The study tested different combinations of nanobodies as capture and detection antibodies to achieve optimal assay performance but did not utilize advanced methods such as epitope binning or structural analysis to precisely identify the binding sites of each nanobody on the NS1 antigen. While the ELISA was optimized for sensitivity and specificity, the lack of detailed epitope identification is a methodological limitation that could be addressed in future work to further enhance assay design (Page 14).

2. The Methods section does not provide extensive details on the use of controls or replicates in some assays, which may affect the reproducibility and robustness of the results (Page 2).

3. The introduction briefly mentions NS1 as a biomarker but does not elaborate on its structural forms, secretion dynamics, or variability among the four DENV serotypes, all of which are important for understanding the challenges in developing broadly reactive diagnostic tools (Page 6).

4. The article does not provide detailed information regarding the use of transfected Vero cells for screening. Although the Methods mention the use of recombinant protein produced in transfected cells for nanobody selection, there is no thorough explanation of the screening process involving these cells. This lack of detail may affect reproducibility and understanding of how well the selected nanobodies recognize the native NS1 protein in a cellular context.

Reviewer #2: Yes to all the questions.

Reviewer #3: - Overall, a logical flow was followed for generation and selection of Nb, and for the selection of best performing clones for capture and detection of NS-1

- The NS1 protein used to immunize the llama is produced in insect cells (C6/36), which can represent a limitation because there is no clear consensus in the literature whether the NS1 produced by insect cells has the same characteristics as the one produced by mammalian cells. This should at least be discussed by the authors

- Why was panning done only against NS1 from Dengue-1 and Dengue-2, if the aim was to generate cross-reactive Nb?

- Line 205: why was RFLP done, what is the benefit compared to the sequencing that was done?

- Why do PSR sequence modified Nb have expression and purification issues? Can this be explained?

**Results**

-Does the analysis presented match the analysis plan?

-Are the results clearly and completely presented?

-Are the figures (Tables, Images) of sufficient quality for clarity?

Reviewer #1: 1. Inconsistent Notation: The difference in notation ("Nanobody 48" vs. "Nb66") likely reflects slight inconsistencies in writing, but both refer to specific nanobodies characterized and used in the assays (Line 428).

2. The article lacks detailed schematic diagrams of the ELISA procedures, which could aid in understanding and reproducing the assay, particularly for the complex combined nanobody formats. Including schematic diagrams for each key step throughout the article is highly recommended. Visual aids such as diagrams of the ELISA procedure and the combined Nanobody ELISA (cNb-ELISA) would greatly enhance readers’ understanding by clearly illustrating the workflow, antibody interactions, and assay design. For example:

• A schematic of the standard Nanobody-based ELISA showing capture antibody coating, antigen binding, detection antibody incubation, and signal development would clarify the optimization steps described (Page 14).

• A detailed diagram of the combined Nanobody ELISA, highlighting the use of specific nanobodies as capture and detection antibodies, sample addition, and detection readout, would help explain the assay’s development and validation process ([14,118,654]; [15,36,432]).

• Including flowcharts or stepwise illustrations of the statistical validation methods (ROC curve analysis, cut-off determination) could also aid comprehension of the assay’s diagnostic performance evaluation (Page 15).

3. The hospital’s algorithm used unusually low cutoff values in some tests, which led to false positives. This underscores the importance of carefully defining cutoff thresholds for the cNb-ELISA to strike an appropriate balance between sensitivity and specificity ([6,50-56]).

Reviewer #2: yes to all teh questions

Reviewer #3: - The results are mainly clearly presented and explained with appropriate figures

- In some cases, the concentrations of the mAbs are expressed in nanomolar (nM) units, while in others, they are expressed in nanograms per milliliter (ng/mL). This may not be very clear for the reader.

- The authors determined the cross-reactivity of the nanobodies only against Zika virus. In case of commercialization, it will be helpful to know whether the assay can also be used in regions where other flaviviruses co-circulate

- It seems that no competitive clustering / epitope binning was done for the Nb selected, as this would allow to select pairs of Nb for capture/detection that do not target the same epitopes/regions. Is there a rationale for not including these type of experiments? Maybe this could be discussed in the discussion section

- It is not clear what the benefit of this study / new Nb is compared to previous studies, like the following publication that seems to be more detailed, especially given that the analysis of cross-reactivity goes beyond ZIKA virus, which seems critical for the proposed diagnostic assay

o Shriver-Lake, L.C., Liu, J.L., Zabetakis, D. et al. Selection and Characterization of Anti-Dengue NS1 Single Domain Antibodies. Sci Rep 8, 18086 (2018). https://doi.org/10.1038/s41598-018-35923-1

- Figure 1: overall, most Nb show very poor capture capacity in this assay. What was the background absorbance observed for capture using Nb that are not specific for NS1? Why was this control not added? Comparison with a non-specific Nb would allow to assess the Nb that show very low absorbance are actually specific in capturing the NS1

Line 402: “the fusion of the Fc domain to combine these benefits”. It is not clear why this Fc fusion is done. What do the authors mean with ‘these benefits’.

- The Fc fused Nb clearly is the most potent as capture antibody (figure 2). How can you explain this, as the bivalent Nb also has two binding sites

- Figure 3 shows a dotted line, what does this represent, this should be explained in the figure legend

- The authors use a polyclonal serum as a capture antibody for the final Elisa setup that is used to test the sera. Such an assay based on a polyclonal antibody as the capture antibody, makes it difficult to standardize the assay and bring it to market. What was the rationale for this approach? Why did the authors not use the Nb developed for this capture? Was this solely related to the fact that the Nb based assay could not detect NS1 in clinical samples? Since only one Nb was finally assessed for capture of NS1 in clinical samples, did the authors test other Nb from their initial panel if these could capture NS1 in clinical samples?

**Conclusions**

-Are the conclusions supported by the data presented?

-Are the limitations of analysis clearly described?

-Do the authors discuss how these data can be helpful to advance our understanding of the topic under study?

-Is public health relevance addressed?

Reviewer #1: The manuscript briefly mentions limitations (e.g., lower detection of DENV-4 NS1) but does not thoroughly explore their implications or potential solutions. The limitation regarding DENV-4 detection is acknowledged but not deeply analyzed or addressed.

The writing could better position the study within the broader context of existing diagnostic approaches by comparing the results and methods with other current diagnostic tools. There is limited discussion on how the nanobody-based ELISA compares to other NS1 detection methods or commercial diagnostic kits.

Some sentences are awkwardly constructed or contain minor grammatical errors that may detract from the manuscript's professionalism. For example, "Considering the restrictions of the public health system in accessing commercially available diagnostic kits" could be rephrased for clarity.

Reviewer #2: Yes to all the questions

Reviewer #3: - A general statement is made about the cost of monoclonal antibodies compare to the cost of Nb. It would be good if this could be substantiated by actual numbers and how this would translate to the actual difference in cost per test. Especially given that the Nb also need to be modified with biotin for the test and that the capture antibody is a polyclonal Ab, and that for the detection antibody a Nanobody HRP fusion that is produced in eukaryotic cells is used. The latter two aspects seem to abolish the possible cheaper cost of using Nb.

- One aspect that does not seem to be considered is regulatory aspects / approval of diagnostic devices. Are there actual diagnostic devices on the market based on nanobodies? It would be good if this could be addressed as this would show the value and viability of the idea of making Nb based diagnostic assays for antigens…

- The authors mention using stable HEK-293T cells to express Nanobody-HRP fusions. Why HEK-293T cells to do this, usually these cells are used only for transient expression?

- Line 712: this part is quite speculative, and if statements are to be made about negative samples that might be considered positive, then probably other clinical samples need to be used that are better validated, or these samples need to be thoroughly tested using different assays to be sure if they are positive or negative. In this context, it could have been interesting to compare the results obtained with the ELISA-based kits with PCR experiments to determine if the specificity and sensitivity of the serological and molecular methods can be compared.

- Building on the previous point, it could have been of interest to include in the patient cohort also the samples positive for Dengue IgM, to try to determine the window in which the assay can be used.

- For the future, urine and plasma samples could also be collected, as it appears that NS1 can be detected for a longer period in as it appears that NS1 can be detected for a longer period in these specimens. Maybe this can be discussed?

**Editorial and Data Presentation Modifications?**

Reviewer #1: (No Response)

Reviewer #2: Before discussing the major issues, let me start with a few easy to amend suggestions:

Line 156 Is the recombinant NS1 the intact gene, does it contain a secretion signal?

Line 171: please add the interval of the boosts and/or timing of boosts. The volume was 250 microL, but what was the concentration?

Line 176 what is the source of the anti-lama IgG HRP conjugate? or how was it made? Writing that it was 1:15,000 diluted is not very helpful if we ignore the original source an/or activity.

Line 186: You mention 'counter selection steps' but it is not clear what these are or how they were performed.

Line 199: "Plates were blocked with PBST" normally BSA, casein, or protein free blocking agents are used. Only PBST? that is strange and would not be considered as blocking residual protein binding sites.

Line 287: "at specific intervals" this is everything except "specific" or informative.

Line 378: it is strange (not-standard) to use equine serum as blocking agent. At what dilution was it used? Presumably this contains ca 20-50 mg Serum albumin/ml. So was the final concentration of HSA around 1% or 3% (comparable to BSA?) then why doesn't it give the same result as BSA?

Major comments:

Thus, it seems that the Nb as NS1 capturing agent is not working efficiently with clinical serum samples, possibly due to the presence of host IgGs of the infected patient. Therefore the capturing Nb was substituted with polyclonal rabbit anti-NS1 serum. Hoever, this is not a sustainable source of antibodies. So, how well can this polyclonal reagent be upscaled to be used in the clinic over a longer period, and will a new batch from another rabbit provide reproducible and comparable results?

The discussion is much too long and repetitive: for example the widespread and increasing frequency of occurrence of Dengue infections has been mentioned in the Introduction section. There eis no need to repeat it in the discussion.

Also the results were clear enough, so that there is no need to repeat the set-ups that didn't work (plastic binder peptides).

The authors mention that it the commercialised diagnostic Dengue kits are expensive and not that easy to obtain in South America. Therefore, this in-house system is claimed to be easier accessible and more economic for the clinical labs. Now, clincal labs are not keen to change the reagents (or provider of their reagents) as it would mean that they should repeat all or most of their previous samples, to guarantee that the same results will be obtained. Therefore, if you want to convince clinical labs to change you absolutely need to demonstrate the sensitivity and specificity of your protocol or kit with that of the commercial widespread kits using exactly the same well documented clinical samples. Without this information nobody will ever try this new set of reagents. (and also they need to be ensured that the reagents will remain available over longer periods (decades)).

Reviewer #3: (No Response)

**Summary and General Comments**

Reviewer #1: (No Response)

Reviewer #2: A new type of ELISA was developed to screen clinical samples on the presence of Dengue NS1. The authors developed a polyclonal rabbit anti-NS1 IgG as capturing agent and used an anti-NS1 Nb conjugated to HRP as detecting agent.

A few details in the methodology are missing and should be added. The discussion should be shortened.

The experimental work was well explained and the authors assessed the perfomance of a number of Nb modifications (plastic binder peptides, IgG reconstruction, biotinylation).

Apparently only the use of rabbit polyclonal anti-NS1 as capturing agent is working with samples from putative infected patients. This might affect the sustainability of the protocol, if it will be used on a larger scale. So the authors aer invited to discuss, how they would be able to cope with this issue.

Reviewer #3: - The paper is overall well written. The experimental setup is straightforward, and the authors provide detailed explanations for the reasons behind the experiments described in the paper.

- The approach, based on nanobodies instead of monoclonal antibodies, is innovative and has the potential to find a place on the market.

- Several aspects related to the methods used and the results are not sufficiently clear and need further clarification

- Conclusions and some statements are not always convincingly supported by own data and/or existing litterature

PLOS authors have the option to publish the peer review history of their article (what does this mean? ). If published, this will include your full peer review and any attached files.

**Do you want your identity to be public for this peer review?** For information about this choice, including consent withdrawal, please see our Privacy Policy .

Reviewer #1: No

Reviewer #2: No

Reviewer #3: No

**Figure resubmission:**

**Reproducibility:**



---

## [Editor Report · Decision Letter 1]

18 Oct 2025

Dear Dr Ibañez,

We are pleased to inform you that your manuscript 'Engineered Nanobodies for early and accurate diagnosis of dengue virus infection' has been provisionally accepted for publication in PLOS Neglected Tropical Diseases.

Best regards,

Guy Caljon

Academic Editor

David Safronetz

Section Editor

Shaden Kamhawi

co-Editor-in-Chief

Paul Brindley

co-Editor-in-Chief

Thank you for addressing the major comments raised and for performing the additional experiments within the limits of what was possible logistically. The limitation of the polyclonal antibody as capture reagent remains, but information about the capturing capacity of monovalent and Fc-fused Nbs has now been included. Absence of cross-reactivity with other viruses is now also better documented.

A few minor comments:

Fig S8&9, panels A&C: adjust the position of annotation "NS1-2"

Please correct line 116: the protein "es"

---

## [Editor Report · Acceptance letter]

Dear Dr Ibañez,

We are delighted to inform you that your manuscript, "Engineered Nanobodies for early and accurate diagnosis of dengue virus infection," has been formally accepted for publication in PLOS Neglected Tropical Diseases.

Best regards,

Shaden Kamhawi

co-Editor-in-Chief

Paul Brindley

co-Editor-in-Chief
